# Neighbourhood Environment and Cognitive Vulnerability—A Survey Investigation of Variations Across the Lifespan and Urbanity Levels

**Marica Cassarino** [1,*] , **Eleanor Bantry-White** [2] **and Annalisa Setti** [1]

1   School of Applied Psychology, North Mall Enterprise Centre, University College Cork, T23 TK30 Cork, Ireland; a.setti@ucc.ie
2   School of Applied Social Studies, William Thompson House, Donovan's Road, University College Cork, T12 D726 Cork, Ireland; e.bantrywhite@ucc.ie
*   Correspondence: mcassarino@ucc.ie; Tel.: +353-21-490-4589

**Abstract:** Background: Understanding the impact of local communities on wellbeing is a paramount Sustainable Development Goal. In order to inform people-centred planning interventions, it is important to understand what demographic groups need supportive communities the most and where. This study explored associations between perceived neighbourhood characteristics and cognitive vulnerability, and the moderating role of age and urbanity of the place of residence. Methods: A convenience sample of 224 Irish adults completed a survey assessing cognitive vulnerability and perceptions of neighbourhood qualities, together with sociodemographic and residential information. Correlational analyses explored bivariate associations as well as moderating effects. A subsample (n = 142) provided qualitative accounts of their preferences of neighbourhood characteristics. Results: Regression analyses showed that controlling for sociodemographic factors, higher self-reported neighbourhood pleasantness was associated with lower cognitive vulnerability, particularly in older adults who lived in the most rural and urban areas ($p$ = 0.006). Qualitative accounts suggested urban–rural variations in perceived accessibility and perceptual stressors, and age-related variations in social preferences. Conclusions: Our findings indicate a complex association between neighbourhood characteristics and cognitive wellbeing, highlighting the potential benefits of neighbourhood pleasantness for cognition particularly for older people in very rural or very urbanised places. Implications for research and environmental interventions are discussed.

**Keywords:** cognitive vulnerability; neighbourhood environment; lifespan; urbanity

## 1. Introduction

A key sustainable development goal [1] is developing supportive communities that foster health and wellbeing [2]. In line with this, there has been increasing interest in understanding how the built environment can impact wellbeing [3]; recently established cross-disciplinary research agendas, such as Neuro-Urbanism [4] and Community Gerontology [5], have proposed to look more closely at the psychological and cognitive impact of the lived environment, particularly, the neighbourhood [6–9], as the most proximal form of environmental exposure outside the house [7]. Within these frameworks, there are growing calls for the promotion of neighbourhood liveability through the design of environments that are cognitive-friendly [8,10] and that foster neurodiversity [11]. Theoretical models that link the neighbourhood environment to cognition propose two main possible pathways, one indirect and one direct [8,9]: Characteristics of the community can have an indirect influence on cognition and mental health by providing or limiting opportunities for a physically and socially active lifestyle,

which in turn can have a knock on effect on cognitive wellbeing via cardiovascular health and social stimulation [6,9]. However, recent research has shown an interest in the possibility that the physical or 'contextual' environment may act as a direct source of cognitive stimulation based on the perceptual and cognitive complexity presented, and, thus, on the demands posed on cognitive skills [7,8]. Considering the direct pathway, it is suggested that environments which are under-stimulating (e.g., rural areas) or over-stimulating (urban areas) from a cognitive point of view may be detrimental for cognitive wellbeing [8,12]. A wealth of research using either objective or subjective measures of the environment has demonstrated that physical aspects of the built environment (e.g., walkability) can have an indirect impact on mental health and cognition [13] by influencing lifestyle [14–16]; however, in order to clarify what makes an environment 'cognitive-friendly', a crucial step is to understand how the sociophysical components of the neighbourhood can influence the cognitive wellbeing of an individual based on the level of perceptual and mental stimulation afforded [8,12,17]. A small number of experimental studies that looked at the direct impact of the physical environment on brain processing using wearable electroencephalography integrated with qualitative accounts of subjective experiences with older and younger individuals show brain activity associated with relaxation when sitting/walking in urban areas characterised by the presence of green, quietness, or in familiar places [18–21]. However, these studies do not take into account the potential influence of individual-level or contextual factors. In the present study, we investigated whether neighbourhood characteristics are associated with cognitive characteristics of the individual and with their sensory sensitivity, and whether age and levels of urbanity moderate this association.

Individual characteristics such as sensory sensitivity can modify the restorative effects of green spaces on cognition [22], suggesting that individual differences in cognitive and sensory processing may influence a person's interaction with outdoor spaces. In line with this, in a recent study with older adults, we observed a negative association between sensory/cognitive sensitivity and preferences for walking in crowded outdoor spaces in individuals living in very rural or very urbanised places, but not in areas with intermediate levels of urbanity [17].

Understanding which demographic groups are the most cognitively vulnerable and what environmental characteristics are associated with such vulnerability is thus important to identify how communities can sustain good cognitive health and design environmental interventions or transformations.

A recent systematic review [6] pointed at the need to study potential individual-level moderators, such as age, to better understand the associations between neighbourhood environment and cognition. The cognitive impact of neighbourhood design has been investigated extensively in relation to vulnerable populations such as individuals with dementia [10,23,24], but there is limited evidence of whether the association between neighbourhood characteristics and cognitive wellbeing varies across the lifespan [25]. We know from existing research that availability of resources or infrastructure (e.g., walkability) in the neighbourhood which promote active lifestyles and social integration provide a mental boost for both younger and older groups [26–29]. However, at the neighbourhood level, the role of the physical environment is underexplored, and it is not known whether cognitive benefits are associated with the proximal environment of residence in younger and older people. Presumably older adults live longer in their neighbourhood; therefore, one may argue for an accumulation effect whereby the impact of the neighbourhood design on cognition could be observed more strongly for older rather than younger individuals [9,12]. Thus, a question that needs investigation is whether neighbourhood characteristics are cognitively relevant for older people only, or whether we can observe an association in younger cohorts also.

Considering the contextual level of influence, epidemiological studies have demonstrated urban–rural variations in cognitive health in older populations [9,30], suggesting a direct impact of the contextual aspects of the lived environment on perception and cognitive outcomes [7]. Considering younger groups, natural settings, as opposed to urban places, appear to benefit cognition, especially in terms of attentional functioning [22,31,32]. However, inconsistencies in the positive impact

of nature across the lifespan have been observed [31], and the same may hold true for urban environments. This poses the question of whether levels of urbanisation moderate the association between neighbourhood characteristics and cognition; addressing such question would enable us to pinpoint where environmental interventions are most needed.

The present study aimed to advance our understanding of the association between subjective perceptions of the neighbourhood environment and cognitive, as well as sensory, vulnerability with the ultimate goal to identify which age groups in which residential areas are most in need of urban design interventions to improve cognitive wellbeing. To this end, we investigated two research questions:

1. Whether neighbourhood contextual qualities were associated with variations in cognitive and sensory vulnerability in adults.
2. Whether the associations between neighbourhood characteristics and cognition were moderated by age and level of urbanity of the place of residence [17,33].

## 2. Materials and Methods

### 2.1. Design

In this cross-sectional study, we analysed quantitative and qualitative ratings of neighbourhood characteristics to study associations with cognitive/sensory vulnerability. We also explored variations in these associations in younger and older adults and based on the level of urbanity of the place of residence.

### 2.2. Participants

A sample of 224, community-dwelling Irish people aged ≥18 years (mean age = 52.34, SD = 20.76, age range = 19–95 years; 72.7% female) completed a survey either online or by pen-and-paper. Participants were recruited through convenience and snowball sampling, using university mailing lists, social media platforms, and liaising with community gatekeepers and existing contacts, especially to reach older people (e.g., active retirement groups, walking groups). Data collection took place between October 2017 and September 2018. Completion of the survey lasted on average between 15 and 30 min. All participants read an information sheet and signed a consent form before taking part in the study. The study was conducted in accordance with the Declaration of Helsinki and received ethical approval (code 19/09/2017) in September 2017 from the School of Applied Psychology Ethics Committee at University College Cork (Ireland). Participants received no compensation for taking part in the survey.

### 2.3. Measures

#### 2.3.1. Cognitive Vulnerability

In this study, cognitive vulnerability was measured as cognitive failures and sensory sensitivity. The 25-item Cognitive Failure Questionnaire, or CFQ [34], was employed to assess the frequency of cognitive failures in the previous six months (from 0 "Never" to 4 "Very often"). An unweighted least squares factor analysis with Promax rotation (Kaiser–Meyer–Olkin = 0.85, Bartlett's Test of Sphericity $chi^2(300) = 1636.89$, $p < 0.001$, determinant = 0.000) confirmed a one-factor structure explaining 27% of the variance, as found in previous studies [35]. Although there are no recommended cut-off scores for this tool, studies have indicated good psychometric properties and its usefulness as a measure of attentiveness in everyday life [35]. We created a cognitive failure score by adding up the scores of the 25 items (Cronbach's alpha = 0.88), with higher scores indicating higher frequency of cognitive failures in day-to-day activities.

Sensory sensitivity was assessed through an 11-item questionnaire adapted from the Adult Sensory Profile and Sensory Processing Scale [36,37]; each item measured the individual's level of sensitivity to sensory stimulation coming from the surrounding environment (on a 5-point Likert scale from 1 "Very much unlike me" to 5 "Very much like me"). A factor analysis of the 11 items employing

unweighted least squares with promax rotation (Kaiser–Meyer–Olkin = 0.75, Bartlett's Test of Sphericity *chi²(55)* = 441.36, *p* < 0.001, determinant = 0.13) indicated two factors explaining, respectively, 28.7% and 12.5% of the variance (confirmed by parallel analysis). All items except two loaded onto one of the two factors, with loading ranging between 0.44 and 0.73. However, only one of the two factors with seven items demonstrated moderate reliability (Cronbach's alpha = 0.76), thus the second was not included in the analyses. A composite score was created by adding up the seven items, with a score ranging from 7.00 to 35.00.

### 2.3.2. Neighbourhood Environment

In the quantitative section of the survey, participants were asked to rate the perceived quality of their neighbourhood of residence, considering "neighbourhood" as the area around the house that could be covered by walking for 15 min, as done in other studies [28]. For the purpose of this analysis, two measures of neighbourhood environment were used:

1. Neighbourhood pleasantness, including the perceptions of safety, cleanliness, aesthetic appeal, absence of noise, and absence of crowding;
2. Neighbourhood streetscape, comprised of perceptions of road pavements, lighting, and permeability.

We initially selected neighbourhood qualities based on existing frameworks of built environment factors for physical and mental health [7,38] and a review of the extant evidence on environment and cognition [8,9], looking at qualities that would be most relevant to cognitive processing in terms of being a potential direct source of sensory and cognitive stimulation. Items in the survey included: streetscape qualities, comprising quality of road pavements, road lighting, and connectivity between roads; perceptual stressors including noise, crowding (from people), and traffic; overall environmental qualities of safety, cleanliness, and aesthetic appeal. Participants rated statements about these neighbourhood characteristics on a Likert scale from 1 (Not at all like my neighbourhood) to 5 (Very much like my neighbourhood). An exploratory factor analysis of the nine items employing unweighted least squares (Kaiser–Meyer–Olkin = 0.71, Bartlett's Test of Sphericity *chi²(36)* = 752.96, *p* < 0.001, determinant = 0.03) indicated two factors explaining, respectively, 32.1% and 28.7% of the variance (confirmed by parallel analysis). In the confirmatory factor analysis using principal axial factoring with Promax rotation, all items except perceptions of traffic showed loadings ranging from 0.53 to 0.78 onto one of the two factors. Both factors demonstrated moderate reliability with a Cronbach's alpha, respectively, of 0.76 (Neighbourhood pleasantness) and 0.79 (Neighbourhood streetscape). The composite scores were calculated as an average of the scores in each item, with scores ranging from 1 (Not at all like my neighbourhood) to 5 (Very much like my neighbourhood).

Participants were also asked in the survey to mention an aspect of their neighbourhood that they liked the most and the aspect that they liked the least, and a subsample (*N* = 142, mean age = 42.59, SD = 18.16, 78.01% female) answered these questions. These two items were coded using a framework of six neighbourhood qualities: (1) Streetscape infrastructure: pavements, lighting, road quality; (2) Perceptual/sensory stressors: noise, clutter, quietness, crowding, physical hazards, pollution; (3) Pleasantness: safety, cleanliness, aesthetic appeal, housing or architectural quality, litter, scenery; (4) Natural elements: green spaces, blue spaces, weather, temperature; (5) Accessibility: close to amenities/services, transport options, walkability, permeability, parking; (6) Social aspects: sense of community, support, social network, familiarity, affordability, tourists. These six dimensions were derived from the two composite measures of neighbourhood environment that emerged in the factor analysis and from a previous study we carried out on preferences for outdoor spaces [17]; elements such as nature or social aspects had not been included in the original items of neighbourhood ratings, but emerged as important to our participants.

Participants had the option to indicate the location or address of their neighbourhood in the survey for auditing purposes; however, only 17% of participants provided this information, which in some

cases was too generic to allow for an appropriate identification. For this reason, the geographical location of the neighbourhood was not included in our analyses.

### 2.3.3. Moderators

Participants' age and level of urbanity of the place of residence were explored as potential moderators of the association between neighbourhood characteristics and cognitive vulnerability. Respondents indicated their age in years; for the purpose of this analysis, participants were divided into three groups based on tertiles of age, so to have a balanced number of participants as well as categories that would reflect three generational groups (18–41 years old; 42–65 years old; 66+). Level of urbanity of the neighbourhood was classified as an ordinal scale of increasing levels of urbanity along five categories: countryside, village, town, city suburbs, inner city. We adopted these categories based on Irish Census categorisations, and further details are provided elsewhere [17].

### 2.3.4. Covariates

Analyses controlled for participants' gender (male, female), their level of education (below third level or degree vs. third level or higher) as a proxy of socioeconomic status, their self-rated health (Likert scale from 1 "Poor" to 5 "Excellent"), and social network (measured as the presence of friends and/or relatives in the neighbourhood: 1—no friends or family in the neighbourhood; 2—friends or family; 3—friends and family). Lastly, we controlled for duration of residence, categorised as 1 "Less than one year", 2 "1 to 5 years", 3 "5 to 10 years", 4 "10–20 years", and 5 "Over 20 years". We asked participants to indicate their occupational status; however, we did not include this measure in the analyses as working or not was very much age-related (i.e., youngest and middle-aged more likely to be working, whereas oldest participants were more likely to be unemployed because retired), *chi$^2$(4)* = 132.38, *p* < 0.001, *Cramer's V* = 0.79.

### 2.4. Data Analysis

For quantitative data, descriptive statistics for the whole sample were presented as mean and standard deviation (SD) for continuous variables; median (Md) and interquartile range (IQR) for ordinal variables; frequencies and percentages for categorical measures. The two outcome measures (CFQ and SS) showed violation of normality (Shapiro–Wilk test *p* = 0.02 for CFQ and *p* = 0.04 for SS), demonstrated absence of a linear relationship with the two measurements of neighbourhood environment, and demonstrated larger variance than mean when stratified by age group or urbanity level; based on these diagnostics, both variables were treated as count variables with overdispersion.

Pairwise nonparametric Spearman's correlations were conducted between the two measures of cognitive vulnerability (CFQ and SS), as well as with the measures of neighbourhood environment and urbanity levels. Comparisons between age groups were carried out using the Kruskal–Wallis test for nonparametric comparisons, with effect size calculated using the Kruskal–Wallis eta-square estimate [39]. Interactions between each of the neighbourhood environment measures, age, and urbanity were analysed using a negative binomial regression model with incidence rate ratios (IRRs), controlling for gender, education, self-rated health, and neighbourhood social network. Four models were developed—two for cognitive failures (CFQ), with each including one of the neighbourhood environment measures, and two for sensory sensitivity (SS). Model fit was estimated using likelihood ratio Chi-square. To account for the two different regression models by two outcome measures, a Bonferroni adjustment was employed to test significant interactions, with adjusted *p* = 0.05/4 = 0.013.

A subsample analysis of qualitative neighbourhood ratings (the most vs. least favourite aspect of the neighbourhood) was carried out by employing summative content analysis [40]. The research team developed the 6-dimension framework described above (see Neighbourhood environment section) and one researcher (MC) used the framework to code the positive and negative ratings: this was done by assigning a numerical code from 1 to 6 to each rating based on keywords. Another researcher (EBW) reviewed the codes to ensure appropriate coding. The qualitative items were then used as

two categorical variables to carry out a quantitative exploratory analysis of association with cognitive measures, age, and urbanity.

An a priori power calculation using G*Power for a linear multiple regression with three predictors (neighbourhood environment, age, and urbanity level), a medium effect size $f^2 = 0.15$ [41], alpha = 0.05, and power = 0.95 to estimate a sample size of 119 participants.

Statistical analyses for quantitative data were conducted using the statistical packages IBM SPSS v.25 [42] and STATA MP v.12 (StataCorp LP, Texas).

## 3. Results

### 3.1. Sample Characteristics

In this sample, participants reported an average Cognitive Failures score of 33.57 (SD = 13.41) on a range from 0 to 100, and an average Sensory Sensitivity level of 19.80 (SD = 5.92) on a range 7–35. Neighbourhood overall quality had a mean of 4.00 (SD = 0.81, range 1–5) and neighbourhood streetscape had a mean of 3.31 (SD = 1.21, range = 1–5). Age groups were organised as follows: n = 73 (33.95%) in the youngest group, n = 73 (33.95%) in the middle-aged group, and n = 69 (32.09) in the older group. Over a third of respondents (38.3%) reported living in city suburbs, followed by 28.8% in the countryside, 13.5% in villages, 13.1% in inner city, and 10.8% in towns. Overall, the sample was highly educated, with 136 (61.3%) participants reporting third or higher level of education; participants reported overall very good health (median = 4.00, IQR = 1.00). Most participants reported having friends or family in their neighbourhood (85.3%). In terms of duration of residence, 40% of participants reported to have lived in their current neighbourhood of residence for more than 20 years, 24% between 10 and 20 years, 7% between 5 and 10 years, 16% between 1 and 5 years, and 13% less than one year.

### 3.2. Bivariate Analyses

Spearman's correlations (Table 1) showed that there were no significant associations between either measure of neighbourhood environment and any measure of cognitive vulnerability or health. On the other hand, cognitive failures and sensory sensitivity were positively correlated with each other (*rho* = 0.29, *p* < 0.001). A statistically significant negative correlation between levels of urbanity and neighbourhood pleasantness (*rho* = −0.46, *p* < 0.001) indicated that participants in more urbanised areas provided poorer ratings of neighbourhood pleasantness; on the contrary, increased levels of urbanity had a moderate positive correlation with ratings of streetscape quality (*rho* = 0.41, *p* < 0.001). No associations between urbanity and the two measures of cognitive vulnerability were observed.

**Table 1.** Spearman's correlations.

| Measure | 2 | 3 | 4 | 5 |
|---|---|---|---|---|
| 1. Neighbourhood pleasantness | 0.05 | −0.12 | −0.02 | −0.46 *** |
| 2. Neighbourhood streetscape | | −0.08 | 0.02 | 0.41 *** |
| 3. Cognitive Failures | | | 0.29 *** | 0.06 |
| 4. Sensory Sensitivity | | | | 0.03 |
| 5. Urbanity level | | | | |

Statistical significance is shown as * *p* < 0.05, ** *p* < 0.01, *** *p* < 0.001.

Comparisons by age group are presented in Table 2. These showed that, while the three age groups did not differ significantly in terms of cognitive vulnerability or neighbourhood streetscape, ratings of neighbourhood pleasantness were significantly more negative in the youngest group,

although the differences were of small size. Notably, there were significantly more people in the younger than the other two age groups residing in the inner city and, conversely, significantly more older people in the countryside, although in all three age groups, over a third of respondents lived in city suburbs.

**Table 2.** Sample characteristics by age group.

| | Age Group | | | |
| --- | --- | --- | --- | --- |
| **Measure** | **Youngest** | **Middle-Aged** | **Older** | *p*-**Value** **(Effect Size)** |
| Cognitive Failures, Md (IQR) | 35.0 (17.0) | 31.0 (18.0) | 33.0 (15.0) | 0.18 |
| Sensory Sensitivity, Md (IQR) | 21.0 (9.0) | 19.0 (7.0) | 20 (7.0) | 0.21 |
| Neighbourhood Pleasantness, Md (IQR) | 3.8 (1.2) | 4.2 (1.2) | 4.4 (0.8) | 0.0004 (0.06) |
| Neighbourhood Streetscape, Md (IQR) | 3.3 (1.7) | 3.3 (1.7) | 4.0 (2.3) | 0.04 (0.01) |
| Urbanity level, n (%) | | | | 0.02 (0.21) |
| Countryside | 13 (17.8) | 21 (29.2) | 17 (24.6) | |
| Village | 7 (9.6) | 11 (15.3) | 11 (15.9) | |
| Town | 8 (10.9) | 9 (12.5) | 7 (10.2) | |
| City suburbs | 26 (35.6) | 25 (34.7) | 31 (44.9) | |
| Inner city | 19 (26.1) | 3 (8.3) | 3 (4.3) | |

IQR—Interquartile range; Md—Median. *p*-values are shown for Kruskal–Wallis test of differences across age groups for ordinal variables and chi-square test for categorical variables. Effect size for significant effects refer to Kruskal–Wallis eta2 for ordinal variables and Cramer's V for categorical variables.

The three age groups did not differ significantly in terms of gender ($chi^2(4) = 4.62$, $p = 0.33$) or health status ($chi^2(8) = 14.27$, $p = 0.07$), but younger people were more likely to have higher levels of educational attainment than people in the middle-aged or older groups ($chi^2(4) = 54.24$, $p < 0.001$, Cramer's $V = 0.35$), while older people were more likely than younger groups to have lived in their neighbourhood for longer ($chi^2(8) = 82.99$, $p < 0.001$, Cramer's $V = 0.44$) and to report a larger social network ($chi^2(4) = 27.56$, $p < 0.001$, Cramer's $V = 0.26$).

Considering levels of urbanity, we found no statistically significant variations in Cognitive Failures (Kruskal–Wallis $H(4) = 2.38$, $p = 0.66$) or Sensory Sensitivity (Kruskal–Wallis $H(4) = 1.89$, $p = 0.75$), but significant differences were observed for both measures of neighbourhood environment: Neighbourhood pleasantness was rated higher in less urbanised areas (Kruskal–Wallis $H(4) = 46.15$, $p < 0.001$), whereas neighbourhood streetscape received more positive ratings in more urbanised areas (Kruskal–Wallis $H(4) = 79.07$, $p < 0.001$). In terms of covariates, there were no significant variations by level of urbanity for gender ($chi^2(4) = 4.63$, $p = 0.33$), health status ($chi^2(16) = 10.13$, $p = 0.86$), or education ($chi^2(8) = 7.08$, $p = 0.53$); however, people in towns were more likely than other groups to have both family and friends living in their neighbourhood ($chi^2(8) = 18.27$, $p = 0.02$, *Cramer's V = 0.29*).

### 3.3. Interactions of Neighbourhood Environment, Age, and Urbanity

Negative binomial regression analyses were carried out separately for cognitive failures and sensory sensitivity, each with two separate models including either neighbourhood pleasantness or streetscape (given the absence of correlation between the two measures). The tables with the results of the regressions are presented in Supplementary File 1 (Tables S1–S4). Here, we present the main findings on the interactions.

### 3.3.1. Cognitive Failures

For cognitive failures (CFQ), when using neighbourhood pleasantness as a predictor and controlling for all covariates, a statistically significant three-way interaction was observed with age and urbanity level, $chi^2(8) = 21.51$, $p = 0.006$. As presented in Figure 1, when compared with younger people, older people showed different patterns of cognitive failures based on where they lived and their ratings of neighbourhood qualities: Older people in the countryside or inner city reported lower cognitive

failures with increasing ratings of neighbourhood pleasantness, whereas those in intermediate levels of urbanity showed a negligent amount of variation in cognitive vulnerability associated with ratings of neighbourhood pleasantness (using countryside as the reference, older people in villages $IRR = 14.48$, $p < 0.001$, towns $IRR = 2.85$, $p = 0.09$, city suburbs $IRR = 3.47$, $p = 0.008$, inner city $IRR = 1.62$, $p = 0.36$). A similar pattern, although of smaller size, was seen for middle-aged participants (villages $IRR = 3.98$, $p = 0.008$, towns $IRR = 1.38$, $p = 0.45$, city suburbs $IRR = 2.23$, $p = 0.03$, inner city $IRR = 0.03$, $p = 0.05$).

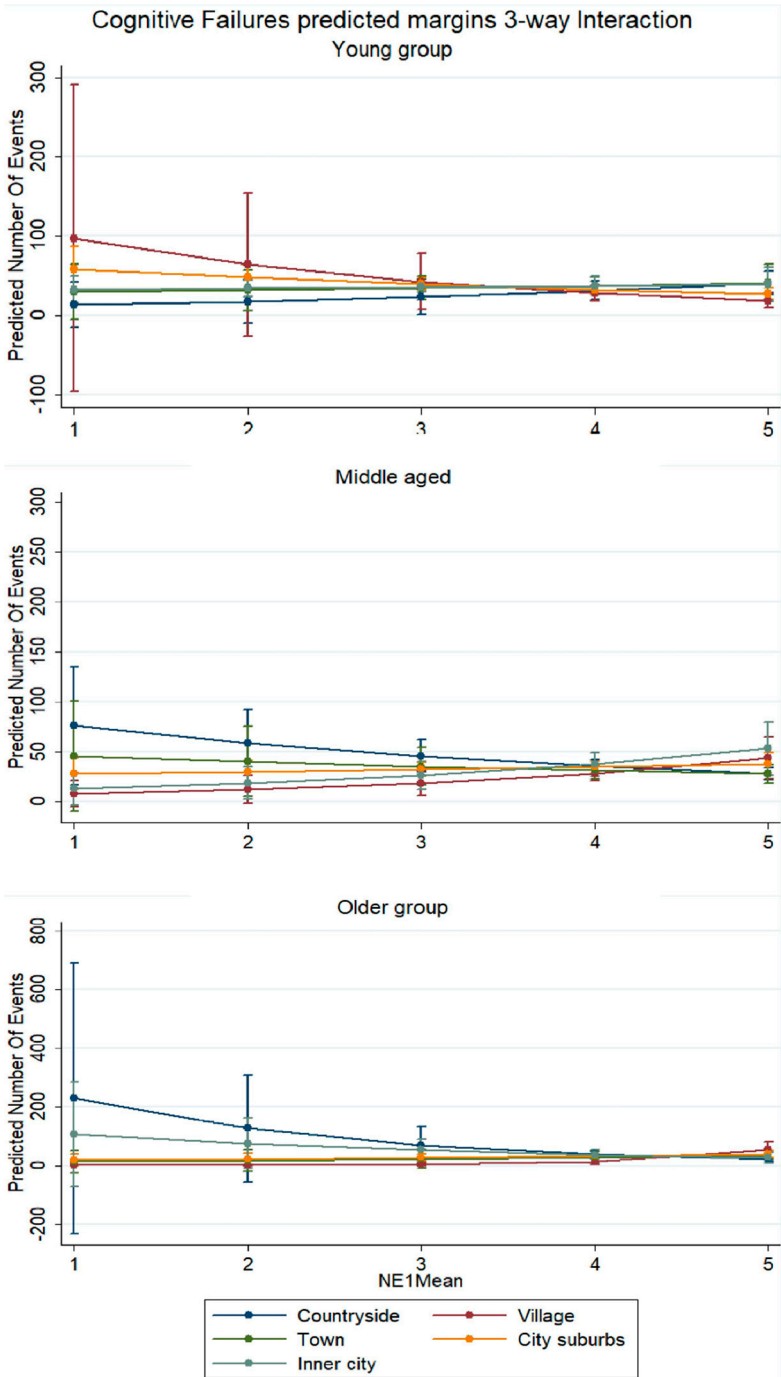

**Figure 1.** Three-way interaction on cognitive failures for neighbourhood pleasantness, age, and urbanity level. Error bars represent 95% confidence interval.

When looking at the effect of neighbourhood streetscape on cognitive failures controlling for covariates, the three-way interaction was not significant (*chi²(8)* = 13.62, *p* = 0.09), and two-way interactions did not reach the significance threshold. A statistically significant effect of age was found (*chi²(2)* =12.83, *p* = 0.002), whereby, compared to the young group, statistically significant lower rates of sensory sensitivity were observed in the middle-aged group (*IRR* = 0.51, *p* = 0.01) and to a larger extent, in the older group (*IRR* = 0.25, *p* = 0.001).

### 3.3.2. Sensory Sensitivity

For sensory sensitivity, when controlling for all covariates, no significant interactions or main effects were noted when considering neighbourhood pleasantness. When using neighbourhood streetscape as a predictor, we observed no statistically significant interactions or main effects (see Supplementary File 1).

### 3.4. Subsample Analysis of Qualitative Ratings

A total of 142 participants (63.39% of the total sample) indicated in the survey the aspect of their neighbourhood that they liked the most and the one they liked the least. The age distribution of the subsample was: n = 72 (51.43%) in the young group, n = 51 (36.43%) in the middle-aged group, n = 17 (12.14%) in the older group. The distribution of urbanity was as follows: n = 24 (16.90%) in the countryside, n = 20 (14.08%) in villages, n = 23 (16.20%) in towns, n = 51 (35.92%) in city suburbs, and n = 24 (16.90%) in inner city. A distribution of positive and negative neighbourhood ratings is presented in Table 3. Over a third of participants (36.62%) named an element related to accessibility when asked about the thing they liked the most about their neighbourhood; these included close distance to services or amenities, as well as public transport. This was followed by social aspects (sense of community, friends living nearby), rated as positive by 22.54% of the sample, and by absence of perceptual stressors (particularly quietness and absence of traffic) mentioned by 18.31% of respondents. Road lighting was mentioned as a positive aspect only by two participants. The two observations were excluded from subsequent bivariate analyses.

**Table 3.** Qualitative neighbourhood ratings.

| Dimension | Positive Rating, n (%) | Negative Rating, n (%) |
|---|---|---|
| Streetscape | 2 (1.41) [1] | 19 (14.73) |
| Perceptual stressors | 26 (18.31) | 41 (31.78) |
| Pleasantness | 12 (8.45) | 23 (17.83) |
| Natural elements | 18 (12.68) | 8 (6.20) |
| Accessibility | 52 (36.62) | 24 (18.60) |
| Social aspects | 32 (22.54) | 14 (10.85) |

[1] Excluded from bivariate analyses due to limited variability.

On the other hand, 31.78% of people mentioned perceptual stressors (such as noise or traffic) as a negative aspect of their neighbourhood, followed by accessibility issues (18.60%, lack of transport or easy access to services or amenities) and issues with pleasantness (17.83%, lack of scenery or litter).

There were no statistically significant age-related variations either for positive ratings (*chi²(8)* = 14.47, *p* = 0.07), although older people (47.06%) valued social aspects of their neighbourhood (sense of community, positive relationships), whereas the majority of middle-aged and young respondents (respectively, 32.65% and 44.44%) valued mainly accessibility aspects of their local area (access to work, services, or amenities). No statistically significant age-related variations were found for negative ratings (*chi²(10)* = 6.28, *p* = 0.79).

Statistically significant associations were observed between neighbourhood ratings and urbanity levels (positive ratings: *chi²(16)* = 53.34, *p* < 0.001; negative ratings: *chi²(20)* = 31.86, *p* = 0.05). As it can be seen in Figure 2, accessibility was rated as a strong positive aspect in urbanised areas, but as a

negative aspect in less urbanised areas; on the other hand, perceptual stressors such as noise and traffic constituted positive aspects of more rural places and negative aspects of urban areas.

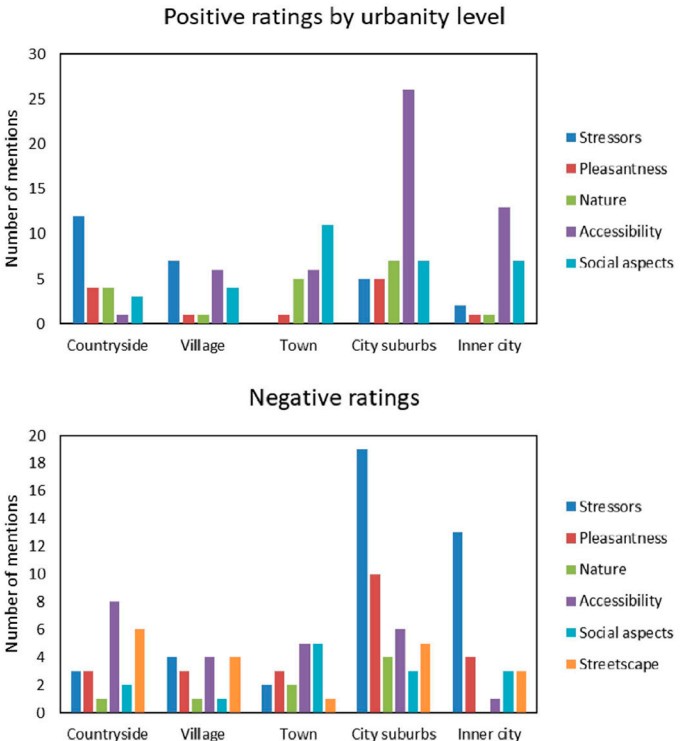

**Figure 2.** Neighbourhood qualitative ratings distributed by urbanity level.

There were no statistically significant associations between neighbourhood ratings and either cognitive failures (positive ratings: Kruskal–Wallis $H(4) = 5.06$, $p = 0.28$; negative ratings: Kruskal–Wallis $H(5) = 1.69$, $p = 0.89$) or sensory sensitivity (positive ratings: Kruskal–Wallis $H(4) = 2.61$, $p = 0.63$; negative ratings: Kruskal–Wallis $H(5) = 0.97$, $p = 0.96$). No interaction analyses were carried out for these ratings, as they would be underpowered by the limited number of observations in each subcategory.

## 4. Discussion

In this cross-sectional study, we aimed to test associations between perceived qualities of the neighbourhood of residence (pleasantness and streetscape) and measures of cognitive vulnerability in a sample of community-dwelling adults, looking at the potential moderation of age and level of urbanity of the place of residence.

We observed no statistically significant independent associations of neighbourhood pleasantness or streetscape with either measure of cognitive vulnerability. Similarly, our subsample analysis found no associations between positive or negative ratings of neighbourhood characteristics and cognitive measures. However, when looking at interactions controlling for demographic, social, and residential covariates, the association between cognitive failures (but not sensory sensitivity) with neighbourhood pleasantness (but not with neighbourhood streetscape) varied significantly based on age and urbanity level. For the oldest group in our sample (when compared to the youngest group), and to a smaller extent middle-aged people, those living in the countryside or in inner city areas were more likely to report lower rates of cognitive failures if rating their neighbourhood as more pleasant, whereas this variation was not observed in older people residing in areas with intermediate levels of urbanisation. Bivariate analyses revealed that overall, participants living in more urbanised areas rated their neighbourhood as less pleasant than those living in more rural areas; also, younger respondents in this study, who were more likely to live in the most urbanised areas, rated their neighbourhoods as less

pleasant than middle-aged or older people; however, older people and, to a smaller extent, middle-aged people in the countryside and inner city had a similar pattern of decreased cognitive vulnerability with increasing neighbourhood pleasantness. This could indicate that neighbourhood characteristics may not determine wellbeing outcomes equally depending on where a person lives, with the most rural and most urbanised areas potentially making a person more susceptible to the perceived levels of neighbourhood safety, cleanliness, aesthetics, noise, or crowding. While the absence of overall age-related variations in cognitive vulnerability, particularly cognitive failures, resonates with the evidence outlined in a recent systematic review of healthy populations [43], these findings support the idea that both age and urbanity levels contribute to influence the association between neighbourhood quality and cognitive vulnerability, revealing for this sample that neighbourhood pleasantness (i.e., how safe, clean, aesthetically appealing and quiet the area of residence is perceived) can contribute to lower rates of cognitive failures, particularly in older people who live in the most rural or urbanised areas. This finding is in line with evidence of interactions between the level of urbanity, neighbourhood perceptions, and quality of life in older people [44], as well as age-related variations in the association between neighbourhood characteristics and wellbeing [25]. Furthermore, as in our sample older people had lived significantly longer in the same neighbourhood compared to younger groups, our findings support the idea that neighbourhood characteristics may be particularly important to older people from a cognitive point of view because of a potential accumulation effect linked to living longer in the same place [8]. However, the present study adds an understanding of these variations with regards to cognitive wellbeing and in a lifespan perspective, confirming a stronger effect in older people rather than younger groups; this gives context to our previous study (conducted on the older group only) where we found that, compared to older people living in areas with intermediate urbanity level, those residing in rural or very urbanised areas who were more cognitively vulnerable were also more selective of outdoor spaces for walking based on the amount of stimulation afforded [17]. Interestingly, the present findings confirm growing epidemiological evidence of a nonlinear association between urbanity and cognitive health in older populations, especially when considering individual-level moderators [6,12,45,46]. Here, broad contextual factors such as urbanity level and neighbourhood-level characteristics seem to operate as two interacting factors for cognitive vulnerability; however, whether there is a cumulative cognitive load of the two (i.e., urbanity and neighbourhood pleasantness provide two different forms of cognitive load) remains to be established.

While we found moderating effects for neighbourhood pleasantness, neighbourhood streetscape was not associated with cognitive measures. It might be that streetscape characteristics were not relevant to our cognitive outcomes for this sample, although other studies have found significant links between street layout and cognitive health in older people [24].

Lastly, in the subsample analysis, we gained further insights on variations in perceived neighbourhood qualities by age and urbanity level. Older people appeared to value more social aspects, whereas younger groups valued accessibility. Considering urbanity levels, the most frequently mentioned dimensions in both positive and negative ratings were accessibility (as a positive aspect of urban areas but negative in rural areas) and the presence/absence of stressors (a positive in rural areas but a negative aspect for urban dwellers). Previous studies have noted the importance attached by older people to social dimensions of their place of residence and outdoor places that they choose to use [17,47]; as social aspects were not investigated in our quantitative questions on neighbourhood characteristics, further investigation of this aspect is warranted. Overall, our findings on variations in accessibility across urban and rural neighbourhoods support the extensive literature on walkability [48] and the negative impact of stressors on health has been demonstrated in urban studies [49]. As these factors can act as mediators between environmental characteristics and cognitive wellbeing [8], they further support the importance of thinking about environmental interventions in light of these contextual features such as urbanity level. As we observed no variations in these ratings across age groups, and we found no clear variations related to neighbourhood streetscape, which was used in this

study as an index of potential hazards and accessibility in the neighbourhood, it is possible that these issues may be relevant to wellbeing independent of age. However, in this study, we found no links between these dimensions and cognitive measures; furthermore, these analyses were only exploratory, given the limited number of participants who provided qualitative ratings.

The present study is one of the few to use a lifespan perspective to look at associations between neighbourhood environment and cognitive wellbeing and combined quantitative and qualitative modes of data collection to gather richer insights on perceptions of neighbourhood characteristics. Nonetheless, the study is not without limitations: Firstly, our convenience sampling, recruitment methods, and self-reported measures mean that we cannot exclude selection bias (e.g., most of our participants were healthy and highly educated) as well as social desirability issues. While the cognitive impact of neighbourhood characteristics has increasingly been demonstrated using both objective and subjective data [19], integrating subjective perceptions of neighbourhood characteristics with an objective assessment and gathering information on the use of the neighbourhood might have provided richer information on variability across neighbourhoods in our sample and enhanced our understanding of contextual factors associated with cognitive vulnerability. Similarly, self-reported measures of cognition should be integrated with an objective assessment of cognition. Furthermore, most participants did not identify geographically the location of their neighbourhoods, limiting our ability to fully capture potential variability across neighbourhoods. The items related to neighbourhood characteristics were informed by the literature and selected as the most relevant to tease out direct associations with cognitive measures [9]. However, the qualitative ratings of neighbourhood characteristics, which were spontaneously provided by participants without prompt, generated aspects of the neighbourhood that had not been included in our quantitative questions (e.g., access to nature and social dimensions); thus, our indexes of neighbourhood pleasantness and streetscape are to be considered as a first attempt to identify cognitive-relevant neighbourhood dimensions, and need further development. We could not explore qualitative ratings interactions with urbanity and age in relation to cognitive measure due to the sample size, which could be addressed in a larger study. We did not include lifestyle measures in this study because the neighbourhood of residence could have an impact independent of how active a person is; however, a potential mediating effect cannot be excluded [50]. The significant interaction observed for cognitive failures supports the study hypothesis that the association between neighbourhood characteristics and cognitive vulnerability is not the same for all individuals, but varies based on their age and their place of residence; however, we cannot rule out the possibility of a spurious finding given the absence of an independent association between neighbourhood pleasantness and cognitive vulnerability. The lack of an independent relationship may be due to the fact that this study included a healthy sample; thus, one recommendation would be to compare the association between cognitively healthy and clinical groups (e.g., older people with mild cognitive impairment). It is also possible that the relationship did not emerge given the medium effect size that we used for our sample calculations, thus warranting further study with a larger sample. Lastly, given the cross-sectional design of the study, we cannot reach conclusions of causality and cannot exclude potential reversed causality (i.e., cognitive characteristics influencing ratings of neighbourhood characteristics); notwithstanding this limitation, our findings point out an important association that deserves further investigation.

Overall, our study stimulates further research not only on the associations between specific neighbourhood characteristics and cognitive wellbeing, but also looking at individual and contextual level moderators. From a theoretical viewpoint, it provides evidence for a direct non-linear association between cognition, perception, and perceived neighbourhood quality, supporting the idea that a positive appraisal of the lived environment is associated with better cognition. This idea should be tested in future work. Our subsample analysis also suggests that elements of safety and social dimensions might be the most important aspects to capitalise upon in areas inhabited mainly by older people, whereas enhancing accessibility and reducing the presence of potential stressors could be applied based on level of urbanity rather than the age profile. Such research may inform more tailored

and cost-effective environmental interventions that take into account comprehensively the demographic profile of the residents as well as broader contextual circumstances.

## 5. Conclusions

In this cross-sectional study, we found a complex association between neighbourhood characteristics and cognitive vulnerability influence both by age and the level of urbanity of the place of residence. The findings of this study thus stimulate an ecological approach to the study of environmental influences on wellbeing with a more comprehensive investigation encompassing potential factors at both the individual and contextual level. The study has also implications for better understanding the factors that influence the types of environmental interventions needed to promote the creation of sustainable communities for wellbeing.

**Supplementary Materials:** The following are available online at http://www.mdpi.com/2071-1050/12/19/7951/s1, Supplementary Table S1: Cognitive failures based on neighbourhood pleasantness, age, and urbanity level. Table S2: Cognitive failures based on neighbourhood streetscape, age, and urbanity level. Table S3: Sensory Sensitivity based on Neighbourhood Pleasantness, age, and urbanity level. Table S4: Sensory Sensitivity based on Neighbourhood Streetscape, age, and urbanity level.

**Author Contributions:** Conceptualization, M.C., E.B.-W. and A.S.; methodology, M.C., E.B.-W. and A.S.; formal analysis, M.C.; investigation, M.C., E.B.-W. and A.S.; resources, M.C., E.B.-W. and A.S.; data curation, M.C.; writing—original draft preparation, M.C.; writing—review and editing, M.C., E.B.-W. and A.S.; visualization, M.C.; supervision, A.S.; project administration, M.C. and A.S.; funding acquisition, E.B.-W. and A.S. All authors have read and agreed to the published version of the manuscript.

**Funding:** This research was partially funded by Seed Funding from the School of Applied Psychology, University College Cork, Ireland.

**Acknowledgments:** None to declare.

**Conflicts of Interest:** The authors declare no conflict of interest. The funders had no role in the design of the study; in the collection, analyses, or interpretation of data; in the writing of the manuscript, or in the decision to publish the results.

**Data Availability:** Data associated with this study can be made available by the authors upon request.

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
