# Peer review of "Neighbourhood Environment and Cognitive Vulnerability—A Survey Investigation of Variations Across the Lifespan and Urbanity Levels"

_sustainability, doi:10.3390/su12197951_

Round 1

Reviewer 1 Report

The author was clear and would further elaborate  Overall it was a very well written article and easy to read and follow. 

Author Response

We thank the Reviewer for the positive feedback and kind words. No revisions were required. 

Reviewer 2 Report

Thank you for the opportunity to review the manuscript entitled, Neighbourhood Environment and Cognitive Vulnerability – A survey investigation of variations across the lifespan and urbanity levels (sustainability-905101) submitted to Sustainability. This study utilizes a convenience sample to examine the relationship between neighborhood perceptions and cognitive vulnerability among Irish community-dwelling adults. While the manuscript is well-written and the study design is acceptable, the study fails to address many of the limitations discussed in Besser et al., 2017. The current study is not novel and adds little to the current literature. Specific comments are listed below.

The abstract should include more information about the sample and information from the regression models.

I think the introduction would benefit from mechanisms that may explain the relationship between neighborhood environment and cognitive vulnerability. The authors should also discuss the assessment of neighborhood environment (perceived vs objective vs administrative data).

How diverse is the sample in terms of SES? Were other measures of SES reported? Can the authors comment on the variability in the participant’s neighborhoods? Since they were recruited from a University mailing list could the participants be clustered in particular neighborhoods? Does the recruitment method limit the variability of participants?  What time period for data collection? How long was the survey? Were participants compensated for their time?

Are there any cutoffs for the cognitive failure questionnaire? Why didn't the entire sample completed the qualitative ratings? Any other measures of SES other than education? Any specific questions on health problems (e.g. dementia, diabetes, cognitive issues)?  Any missingness? How did the authors decide on effect size of 0.25 for the power analysis? Any assessment of home life? Working?

I am surprised that there is no variability in the cognitive vulnerability by age? Is this consistent with other studies? Concerns about validity of the test? For the interaction models, were the bivariate associations statistically significant? Are these spurious findings?

The discussion is well-written and highlights the limitations of the study, many limitations mention in Besser et al 2017. I would disagree that the study is “well-powered.” Other limitations include time spent in the environment (which is also associated with accumulation) and the lack of detail regarding the variability in neighborhood environment.

Author Response

We thank the Reviewer for the constructive feedback. Find below a point-by-point response to the comments. 

R2C1: The abstract should include more information about the sample and information from the regression models.

Response – Thank you for this comment. We have now clarified in the Abstract that this was a “convenience” sample (line 16), that the findings emerged from regression analyses (line 21), and we have included the statistical significance of the 3-way interaction (line 23). We have made some minor changes to other parts of the text (lines 13, 16) accommodate these revisions within the 200-word limit of the abstract.

R2C2: a) I think the introduction would benefit from mechanisms that may explain the relationship between neighborhood environment and cognitive vulnerability. b) The authors should also discuss the assessment of neighborhood environment (perceived vs objective vs administrative data).

Response

  1. a) We have now made more explicit the theoretical underpinnings of the study with regards to the mechanisms linking neighbhourhood characteristics to cognitive vulnerability (lines 40-49):

Theoretical models that link the neighbourhood environment to cognition propose two main possible pathways, one indirect and one direct [8,9]: On one hand, characteristics of the community can have an indirect influence on cognition and mental health by providing or limiting opportunities for a physically and socially active lifestyle, which in turn can have a knock on effect on cognitive wellbeing via cardiovascular health and social stimulation [6,9]. On the other hand, recent research has shown an interest in the possibility that the physical or ‘contextual’ environment may act as a direct source of cognitive stimulation based on the perceptual and cognitive complexity presented, and, thus, on the demands posed on cognitive skills [7,8]. Considering the direct pathways, it is suggested that environments which are under-stimulating (e.g., rural areas) or over-stimulating (urban areas) from a cognitive point of view may be detrimental for cognitive wellbeing [8,12].”

  1. b) We have also made more explicit across the Introduction (lines 50, 57) the point that associations between neighbourhood environment and cognition have been investigated both objectively and subjectively in the literature. We have specified at lines 98-99 that our study looked at “subjective perceptions” of the neighbourhood environment. While we feel that a full discussion of different methods of assessment of the environment is beyond the scope of this paper, we have now acknowledged the lack of an objective assessment of the neighbourhood as a limitation in this study (lines 458-463):

While the cognitive impact of neighbourhood characteristics has increasingly been demonstrated using both objective and subjective data [19], integrating subjective perceptions of neighbourhood characteristics with an objective assessment might have provided a richer information on variability across neighbourhoods in our sample and enhanced our understanding of contextual factors associated with cognitive vulnerability

R2C3: How diverse is the sample in terms of SES? Were other measures of SES reported? Any other measures of SES other than education?

Response - We used educational attainment as a proxy of socio-economic status, and we have now made this clear in the Covariates section (line 202). While we acknowledge that this is a crude proxy of socio-economic status, we opted for this measure in line with the existing literature and also to avoid making the survey too intrusive or difficult to complete, given that participants completed the survey on their own.

As we have now clarified at lines 206 -210, we asked participants to indicate their occupational status, however, we did not include this measure in the analyses as working or not was very much age-related (i.e., youngest and middle aged more likely to be working whereas oldest participants were more likely to be unemployed because retired), chi2(4) = 132.38, p < .001, Cramer’s V = 0.79.

R2C4: Can the authors comment on the variability in the participant’s neighborhoods?

Response - We have now clarified in section 2.3.2 that we asked participants to give us information about the geographical location of their neighbourhood, however, the response rate was very low (187-190): “Participants had the option to indicate the location or address of their neighbourhood in the survey for auditing purposes; however, only 17% of participants provided this information, which in some cases was too generic to allow for an appropriate identification. For this reason, the geographical location of the neighbourhood was not included in our analyses

To give a better sense of variability by neighbourhood, which we clustered in terms of level of urbanity, we have now included some bivariate comparisons for level of urbanity (lines 287-295): “Considering levels of urbanity, we found no statistically significant variations in Cognitive Failures (Kruskal Wallis H(4) = 2.38, p = .66) or Sensory Sensitivity (Kruskal Wallis H(4) = 1.89, p = .75), but significant differences were observed for both measures of neighbourhood environment: Neighbourhood pleasantness was rated higher in less urbanised areas (Kruskal Wallis H(4) = 46.15, p < .001), whereas neighbourhood stressscape received more positive ratings in more urbanised areas (Kruskal Wallis H(4) = 79.07, p < .001). In terms of covariates, there were no significant variations by level of urbanity for gender (chi2(4) = 4.63, p = .33), health status (chi2(16) = 10.13, p = .86), or education (chi2(8) = 7.08, p = .53); however, people in towns were more likely than other groups to have both family and friends living in their neighbourhood (chi2(8) = 18.27, p = .02, Cramer’s V = 0.29).

We agree that this is a limitation of our study, and we have now made this explicit in the limitations (lines 464-466): “most participants did not identify geographically the location of their neighbourhoods, limiting our ability to fully capture potential variability across neighbourhoods

R2C5: Since they were recruited from a University mailing list could the participants be clustered in particular neighborhoods? Does the recruitment method limit the variability of participants?

Response - We have amended the recruitment section to clarify that participants were recruited not only through a University mailing list but also through community links (lines 116-119):

Participants were recruited through convenience and snowball sampling online through University mailing lists, social media platforms, and through community gatekeepers and existing contacts, especially to reach older people (e.g., active retirement groups, walking groups).

We acknowledge that the recruitment methods pose a potential selection bias, as noted in the Limitations section (lines 456-457).

R2C6) What time period for data collection? How long was the survey? Were participants compensated for their time?

Response – We have clarified in the Participants section (lines 119-120) that data collection took place between October 2017 and September 2018, that completion of the survey lasted on average between 15 and 30 minutes (lines 120-121). We have also noted that participants did not receive any compensation for taking part in the survey (lines 124-125).

R2C7: Are there any cutoffs for the cognitive failure questionnaire?

Response – There are no cut-offs for the Cognitive Failures Questionnaire, as this tool is not used as a clinical tool for the identification of cognitive impairment. Studies that have tested the psychometric properties of this tool (e.g., Bridger et al., 2013) recommend this tool as a measure of attentiveness in daily life that is “shown to have excellent psychometric properties that make it suitable for use in both laboratory and field studies as a trait measure of attentiveness in daily life”. We have now included this information and relevant reference in section 2.3.1 (lines 133-135).

R2C8: Why didn't the entire sample completed the qualitative ratings?

Response – We have reworded the relevant text in section 2.3.2 to clarify that all participants were given the option to provide qualitative ratings, but a subsample answered this question (lines 174-176): “Participants were also asked in the survey to mention an aspect of their neighbourhood that they like the most and the aspect that they liked the least, and a subsample of participants (N = 142, mean age = 42.59, SD = 18.16, 78.01% female) answered these questions

R2C9: Any specific questions on health problems (e.g. dementia, diabetes, cognitive issues)?  Any missingness? Any assessment of home life? Working?

Response  - Survey respondents were asked specific questions about their cardiovascular health, including whether they had or not hypertension, experience of heart attack, diabetes, stroke, ministroke or TIA, high cholesterol, any other heart trouble; we also asked about conditions related to mental and emotional health, including Parkinsons’ disease, emotional or psychiatric problems, alcohol or other substance misuse, Alzheimer’s disease, dementia, serious memory impairment. As we note in the sample characteristics, the sample was highly skewer towards healthy individuals, we over 80% of participants indicating to have none of the above conditions, thus, these measures were not included in the regression analyses to avoid over-adjustment. The lack of heterogeneity in health status is likely linked to the sampling methods, and we have noted this in the limitations (line 457): “we cannot exclude selection bias (e.g., most of our participants were healthy and highly educated)

We did not assess home life. We refer the Reviewer to our response to comment R2C3 where we explained our approach to collecting data on working status.

R2C10: How did the authors decide on effect size of 0.25 for the power analysis?

Response – We followed Cohen’s guidelines on effect size. We have now included Cohen’s reference in the Data Analysis section and specified that f = 0.25 is recommended as a medium effect size (line 239).

R2C11: I am surprised that there is no variability in the cognitive vulnerability by age? Is this consistent with other studies? Concerns about validity of the test? For the interaction models, were the bivariate associations statistically significant? Are these spurious findings?

Response – Our finding on no differences in cognitive failures across age is in line with the existing evidence, as noted in a recent systematic review (Carrigan & Bargus, 2016; https://doi.org/10.1016/j.neubiorev.2016.01.010). We have highlighted this in the Discussion section (lines 402-404). We have also included reference to studies that support the psychometric properties of the CFQ in the methods section 2.3.1 (lines 133-135).

With regards to the Reviewer’s comment on bivariate associations related to the interaction models, we find that there were no statistically significant bivariate associations between cognitive failures and either measures of neighbourhood (see correlations in Table 1), age (as indicated in Table 2) or levels of urbanity (included now at lines 287-288). The statistically significant interaction was observed when controlling for covariates, thus limiting the risk of a potential spurious association. However, we acknowledge that the correlational nature of our analyses cannot fully explain potential causal relationships, as we note in the Discussion, lines 476-479.

R2C12: The discussion is well-written and highlights the limitations of the study, many limitations mention in Besser et al 2017. I would disagree that the study is “well-powered.” Other limitations include time spent in the environment (which is also associated with accumulation) and the lack of detail regarding the variability in neighborhood environment.

Response – We thank the Reviewer for this comment. We have now removed the mention on power in the Discussion. We have also acknowledged the lack of an objective assessment of neighbourhood variability and use of the environment as a limitation in this study (lines 458-463):

While the cognitive impact of neighbourhood characteristics has increasingly been demonstrated using both objective and subjective data [19], integrating subjective perceptions of neighbourhood characteristics with an objective assessment and gathering information on the use of the neighbourhood might have provided a richer information on variability across neighbourhoods in our sample and enhanced our understanding of contextual factors associated with cognitive vulnerability

Reviewer 3 Report

This article is devoted to interesting and important issues. Although there are some aspects to be improved:

  • Figures 1 and 2 should be presented in the article or shown in the supplementary files.
  • There is a typo to be corrected – a full stop in the middle of sentence (lines 350-352):

: <…> neighborhood pleasantness (but not with neighborhood streetscape) varied significantly based on age and urbanity level: For the oldest group in our sample (when compared to the youngest group), and to a smaller extent middle-aged people. those living in the countryside or in inner city areas were <..>.

Author Response

We thank the Reviewer for the positive feedback. Find below responses to comments. 

R3C1: Figures 1 and 2 should be presented in the article or shown in the supplementary files.

Response: We are sorry if the Reviewer was not able to access the Figures, which were included in the submission as a zip file. We have now re-included them in the format prescribed by the Journal.

R3C2: There is a typo to be corrected – a full stop in the middle of sentence (lines 350-352):: <…> neighborhood pleasantness (but not with neighborhood streetscape) varied significantly based on age and urbanity level: For the oldest group in our sample (when compared to the youngest group), and to a smaller extent middle-aged people. those living in the countryside or in inner city areas were <..>.

Response: We have now turned the semi-colon into a full-stop at line 389, as suggested by the Reviewer.

Round 2

Reviewer 2 Report

see attached 

Author Response

We thank the Reviewer for the constructive comments and for giving us the opportunity to further improve the manuscript. Please find a point-by-point response hereafter.

R2C1: The second sentence of the abstract does not read well. Some of the language is too casual and repetitive “on one hand,” “on the other hand.”

Response – We have amended the abstract sentence as follows (lines 12-15): “In order to inform people-centred planning interventions, it is important to understand what demographic groups need supportive communities the most and where”.

We have removed “on one hand” and “on the other hand” from the introduction.

R2C2: There is a missing space on line 117.

Response – We amended the sentence to read “Participants were recruited through convenience and snowball sampling, using University mailing lists, social media platforms, and liaising with community gatekeepers and existing contacts” (lines 117-119)

R2C3: The authors should use p < .001 instead of p = .000 and be consistent with the number of significant digits throughout.

Response – Thank you for pointing out these errors. We have amended the p = .000 to p < .001 throughout the paper. We have revised all p-values to ensure that they show two decimal positions, unless they are smaller than .01 (e.g., p = .006).

R2C4: What is the effect size based on for line 239, the majority of studies find a small effect size for neighborhood characteristics.

Response –  We thank the Reviewer for this comment that enabled us to identify the typo in the effect size. This should read 0.15 rather than 0.25. We have now fixed that at line 240; apologies for the confusion. We used a medium effect size as a rule of thumb given the paucity of studies that had tested a research question similar to ours. We acknowledge that adopting a smaller effect size, and thus a larger sample, may hold different results, thus, we have included this observation in the limitations (lines 484-486): “It is also possible that the relationship did not emerge given the medium effect size that we used for our sample calculations, thus warranting further study with a larger sample

R2C5: Tables should be able to stand alone, table 2 needs a footnote for Md (assuming median?).

Response – We have added the footnote to table 2 as recommended. We have also specified in the Data analysis section that median is presented as “Md” (line 213).

R2C6: If there was no correlation between neighborhood pleasantness and neighborhood streetscape, why not include them in the same model and avoid concern of multiple comparisons?

Response – Thank you for this comment. We used two separate models to test the 3-way interaction for each of the neighbourhood measures rather than comparing the main effects of the two measures. While we agree that this forced us to adjust for multiple comparisons, it was felt to be more appropriate to address the research questions for this study.

R2C7: There are no statements that indicate an independent relationship between neighborhood and the outcomes. How can there be moderation without an independent effect? This remains a concern, there is no justification to indicate that this is not a spurious finding?

Response – We have changed the wording of the statement at line 384 of the Discussion from “bivariate associations” to “independent associations”. We have added the following statement to our limitations section (lines 477-484) in response to the Reviewer’s concern: “The significant interaction observed for cognitive failures supports the study hypothesis that the association between neighbourhood characteristics and cognitive vulnerability is not the same for all individuals, but varies based on their age and their place of residence; however, we cannot rule out the possibility of a spurious finding given the absence of an independent association between neighbourhood pleasantness and cognitive vulnerability. The lack of an independent relationship may be due to the fact that this study included a healthy sample, thus, one recommendation would be to compare the association between cognitively healthy and clinical groups (e.g., older people with mild cognitive impairment)”.